# The Cow Paradox—A Scoping Review of Dairy Bovine Welfare in India Using the Five Freedoms

**DOI:** 10.3390/ani15030454

**Published:** 2025-02-06

**Authors:** Chirantana Mathkari

**Affiliations:** Department of Animal Behavior, Ecology, and Conservation, & Anthrozoology, Canisius University, Buffalo, NY 14208, USA; mathkarc@canisius.edu; Tel.: +1-716-888-2775

**Keywords:** dairy cattle, dairy cow, dairy buffalo, cow slaughter ban, five freedoms, animal welfare, India, human–animal relationship

## Abstract

India, the leading producer and consumer of milk, houses nearly a third of the world’s cattle, making the examination of these animals’ welfare essential. The humans’ treatment of an animal is an important determinant of their welfare, and in the case of India, is capable of being shaped by the religiously and financially influenced widespread ban on the slaughter of cows. Against this background, this research aims to evaluate the welfare state of the Indian dairy cattle by analyzing their welfare at various life phases. The results demonstrate an overall compromised welfare state in the animals, followed by a further deterioration in the welfare once the animal is no longer productive. The results signify the role religious values, legislation, and the animal’s economic utility play in the humans’ treatment of the bovines. This study highlights the largely ignored albeit crucial role of the culturally shaped human influences on dairy husbandry and creates grounds for studying human–cattle relationships through interdisciplinary lenses to generate culturally viable solutions to improve cattle welfare and promote a sustainable inter-species coexistence.

## 1. Introduction

India leads the global bovine population charts with 307 million dairy-purpose bovines, accounting for nearly a third of the world’s cattle [1]. The cows and buffaloes together contribute to 97% of the total milk produced in the nation [2]. A ban on the slaughter of cows exists in the majority of the Indian states (Figure 1), the existence of which is intricately interweaved with the disputable yet publicly acknowledged sentiment of the cow’s sacrality, as well as with the animal’s utility value. Largely governed by India’s legislative and economic systems, the slaughter ban plays an imperative role in the welfare of the dairy-purpose cows and buffaloes by influencing their treatment by the nationals.

Although ancient Hindu scriptures substantiate the then common practice of cow slaughter for food, domestic rites, medical treatment, and religious sacrifices, there has been a steady transition towards abandoning cow slaughter in the Hindu religious practices in certain parts of India over the centuries [4]. However, the contemporarily accepted sacrality of the cow is not supported through any of the Hindu scriptures before, during or after this transition [4]. The other dairy bovine of India, the buffalo, although linked with religious significance in the prominent pre-Vedic Hindu scriptures [5], is ironically not regarded holy in the contemporary Indian Hindu ideology, and indeed goes thoroughly unrecognized in India’s current Hindu culture [6]. The legal ban on cow slaughter in a majority of the Indian states, which stems through the acclaimed holiness of the cow and the disregarded religious worth of the buffalo, is therefore paradoxical.

The origins of the cow slaughter ban are deep-rooted in the post-independence socio-political scheme of promoting nationalism. Although the cow’s acclaimed sacredness had been utilized pre-independence as rebellion against the British rule [7], the politically driven intent of using the cow as a Hindu, mother symbol surfaced prominently post-independence. The Hindu right-wing lobbied for the cow’s sacredness to propagate Hinduism and the alleged caste superiority within the religion [8]. The Indian National Congress, the then major political party, desiring to attract the Hindu votes, and to utilize the dairy industry as a tool to establish the newly independent nation’s prowess internationally [9], unanimously voted for the inclusion of the cow slaughter prohibition directive in the Indian Constitution [10].

Article 48, Constitution of India 1950, the cow slaughter prohibition directive from which the state level restrictions and bans governing bovine slaughter are derived, reads:


*“The State shall endeavour to organise agriculture and animal husbandry on modern and scientific lines and shall in particular, take steps for preserving and improving the breeds, and prohibiting the slaughter, of cows and other milch and draught cattle.”*


There exists a certain amount of heterogeneity in the various Article 48-based state laws regarding the slaughter of bovines, indicating an influence of the animal’s economic value in addition to that of religious beliefs surrounding the cow. For example, in Gujarat, a state governed by Bharatiya Janata Party (a political party predominantly backed by the Hindu right-wing) [11], the state legislation Bombay Animal Preservation Act, 1954 implements a complete ban on the slaughter of cows, cow bulls, and bullocks. However, in Uttar Pradesh, one of the largest beef exporters of India [12], the Uttar Pradesh Prevention of Cow Slaughter Act, 1955 permits the slaughter of cow bulls and bullocks that have been certified as fit-for-slaughter by a state governmental authority. The close association of such religious and economic influences is indeed visible in the wording of Article 48, which is drafted to accommodate, alongside the cow’s politicized sacrality, the financial benefits of bovine husbandry.

One of the early and most prominent displays of independent India’s economic outlook towards dairy bovines is the launching of Operation Flood, the first large-scale dairy venture of the nation. This government-led initiative, which ran over two decades (1970–90), made India’s dairy sector organized, increased the number of milch bovines, subsidized the production of dairy products, and boosted the nation’s milk production from 22 million tons in 1970 to 104 million tons in 2000 [13]. By the turn of the century, India became one of the world’s biggest dairy and beef economies [13].

India has consistently ranked number one in milk production since the late 1990s [14], with cows and buffaloes together producing 97% of the total milk [2]. The nation has also been one of the highest beef exporters since the late 2000s [15], indicating the flourishing of its beef industry as a byproduct of its dairy industry. However, the recognition of India’s dairy industry in association with the beef industry severely contrasts with the nation’s anti-cow slaughter sentiment, making the association largely unacknowledged in the nation publicly.

The culturally originated legal binding that prevents dairy farmers from sending the majority of the unproductive cattle for slaughter exerts a significant negative impact, especially on the small-scale dairy farms, which make up the majority of the dairy farms in the nation [16]. Rearing unproductive cattle limits the resources available to provide adequate feed, shelter and veterinary care, compromising the entire herd’s welfare. The strain caused by the inability to slaughter the revered cow, and the economic costs of the animal’s lowered productivity seem to result in the Indian dairy industry resorting to ethically unfit means of dispensing with the animals. The unwanted dairy bovine meets one of the following fates (Figure 2): The bovines, upon serving their economic purpose, are sent to animal shelters, abandoned on the streets, or in the case of the sub-optimally producing animals, sold in animal markets. In the limited states permitting the slaughter of the bovine, the animals may be transported from the farm to the slaughterhouse directly. Bovines abandoned on the streets or sold in markets are eventually transported to a slaughterhouse, either legally or illegally. The transport of cows from shelters to slaughterhouses remained unknown, possibly because of the religious association of several cow shelters in the nation [17,18].

Against this background, I aim to obtain insight into the welfare of the Indian dairy bovines, both cows (*Bos taurus* and *Bos indicus*) and water buffaloes (*Bubalus bubalis*) (the latter henceforth referred to just as buffaloes), by extracting and evaluating the scientific findings on their welfare as obtained through studying it at one or more phases of their lives— in dairy farms, in breeding centers, in animal shelters, on the streets, in transportation, in animal markets, and in the slaughterhouse; followed by a synthesis of the phase-wise analysis to draw holistic conclusions on the animals’ welfare state. To accomplish this, I utilize the Five Freedoms of animal welfare framework (Table 1) which helps to examine an animal’s physical and mental states in relation to their physical as well as social environment [19]. Used widely across nations in legislation, market and farm assurances, and by veterinarians to promote positive welfare [19], the Five Freedoms form a rational and foundational method of measuring the welfare of Indian dairy bovines.

This foundational method was chosen over other more recent frameworks of measuring welfare (e.g., Five Domains framework [20,21], Welfare Quality protocol [22]) as little research has been conducted on understanding the welfare of the Indian dairy bovines, and the Five Freedoms provide an objective and measurable framework to explore a previously unexplored topic such as this. Evidence suggests that this framework has been successfully applied in similar contexts to assess animal welfare [23,24,25], making it a suitable choice for this study.

I hope that this review will help shed light on the animal aspect of this human–bovine relationship and enhance our understanding of the largely unrecognized influence of culture on dairy bovine welfare. I hope that my work will also have some practical significance for the lives of the Indian dairy bovines and their human beneficiaries, as well as for the shaping of culture-specific animal policy and law.

## 2. Materials and Methods

The methodology utilized was that of a scoping review given its appropriateness in systematically reviewing broad research areas as well as its suitability for collectively analyzing the results of studies with clearly variable methodologies [20]. The methodology’s usefulness to identify gaps where evidence from primary methods is lacking and needed, and to target research areas for further review [26,27] further enhanced its suitability for this study.

Peer-reviewed publications pertaining to the measurement of one or more of the Five Freedoms in the Indian dairy bovines were included. Commercial publications were also included to widen the material available to include for review. The timeframe for the included literature was 1950, the year in which Article 48 of the Indian Constitution was implemented, until the year 2024. The end date for included searches was 31 October 2024.

The eligibility criteria were that studies must be either experimental, observational, or generate secondary data, and must scientifically assess the welfare of Indian dairy bovines. Studies assessing the effects of nutritional, environmental (physical and social), management, animal health-related and/or behavioral influences on welfare were included. Studies which assessed the effects of these five influences exclusively on the animal’s production measures (such as milk quantity) were excluded. Additionally, studies which examined the effects of these five influences exclusively on human welfare (such as consumer perception of dairy products, spread of zoonotic diseases to humans), were excluded. Research examining the welfare of the Indian dairy bovines in India as well as in the neighboring countries where the bovines are transported for slaughter (e.g., Bangladesh) [28,29,30,31], was included. Article language was restricted to English given its prevalence in scientific education and research in the country [32]. Although the second official language of India, Hindi faces an inadequacy of scientific terminology, limiting the scientific research published in this language [32,33]. Articles published before the implementation of Article 48 were excluded. No additional restrictions such as the sample size or the quality of the journal were imposed to allow for a comprehensive analysis of the relevant literature. All the above-mentioned criteria for the inclusion and exclusion of articles were defined a priori.

The literature review was conducted from February until August 2024 and again in October 2024. Google Scholar, Scopus, and ScienceDirect databases were utilized for reviewing peer-reviewed scientific articles. Google Scholar is currently the most comprehensive academic search engine [34], Scopus appears to have a predisposition to animal welfare science articles [35], and ScienceDirect retrieves the most relevant articles from the database using a fine-tuned semantic search model [36]. All searches were performed using keyword search terms as required to evaluate the individual freedom of welfare with regards to the purpose of the bovine. For instance, (“India” AND “calf” OR “cow” OR “buffalo” OR “bull” OR “cattle” OR “bovine” AND “welfare” OR “health” OR “behavior” AND “feed” OR “diet” OR “water”) was one of the methods for analyzing the freedom from hunger and thirst. Commercial Indian media publications were reviewed through Newsbank media database’s regional collections. Supplementary searches were conducted over Google to ensure the reviewing of all relevant studies including books, government publications, institutional reports, and PhD theses published in English.

Figure 3 illustrates the procedure performed to select and discard bibliographic sources. In total, 1844 studies were imported for screening, of which 784 were duplicates. From the rest, 344 were deemed irrelevant (did not pertain to the pre-set inclusion criteria) from a title and abstract search, and 532 were excluded following a full text analysis. The following are the results based on the analysis of 184 publications on the welfare of Indian dairy bovines.

Extraction and organization of the literature was performed using Microsoft Excel, version 22H2. The software allowed for the collection of information such as the title, author details, publication year, bovine studied (cow/buffalo), methodology, and results. It additionally allowed for the organization of the data based on the life phase of the bovine and the freedom/s of welfare impacted. The life phases of the bovines’ lives (namely, dairy farms, breeding centers, animal shelters, streets, transportation, animal markets, and slaughterhouse) have been used to structure the findings of each welfare freedom.

## 3. Results

### 3.1. Brief Summary of the Findings

The above-stated findings are derived from the analysis of 184 pertinent articles (those meeting the pre-set inclusion criteria), of which 168 are peer-reviewed and 16 are non-peer reviewed. Among the peer-reviewed literature, 120 articles are based on primary research, while 48 comprise secondary data (For a freedom-wise distribution of the bibliographic references, please refer to Table 2).

Overall, these findings indicate an inability of the Indian dairy bovines to attain the Five Freedoms as a result of inadequate or inappropriate management practices. These undesirable management practices arise from a lack of resources available to and/or a lack of technical know-how in the animal rearers, and can impact multiple freedoms simultaneously. The freedom from hunger and thirst is compromised due to the inferior quality and limited quantity of feed and water available, while the freedom from discomfort is affected by the use of sub-standard infrastructure or make-shift arrangements to house these animals. Lack of accessibility and/or affordability of veterinary care, often coupled with physical discomfort and hunger, deprive the bovines of the freedom of pain, injury, and disease. The freedom to express normal behavior and the freedom from fear and distress are impacted majorly by the lack of attainment of the prior three freedoms, and are commonly found compromised in these bovines. These two latter freedoms, which focus on the mental status of the animal, are also interdependent on each other, such that the loss of one negatively impacts the attainment of the other.

### 3.2. Freedom from Hunger and Thirst

#### 3.2.1. Dairy Farms

The frequent occurrence of dystocia [37,38,39], the duration of which can be prolonged due to the farmer’s hesitance to seek veterinary care [40] and unscientific attempts at resolution [41], hinders the calf’s access to maternal blood supply and thereby, to prenatal nutrition [42]. The small number of calves that survive such prolonged periods of stress are known to exhibit slower suckling reflexes [43], reducing their ability to obtain colostrum from the mother. The majority of the Indian dairies separate the calf from the mother immediately after birth, with suckling permitted only briefly before milking to stimulate milk let down [44,45]. In states that ban the slaughter of cow calves, one in every four male calves is known to die of neglect [40], reasonably due to starvation and thirst. Female calves are reared on milk replacers [46] (p. 269) which can limit the calf’s nutrient availability [47] and is associated with behaviors indicative of hunger [48,49]. With age, calves are transitioned from the replacer to a solid diet; however, a little over half of the Indian dairy farms feed less than half of the desired minimum feed quantity (44 pounds) to their animals [40] and the provision of inferior quality feed is commonplace [50]. Large-scale, organized farms, which are relatively small in number, ensure the provision of adequate feed quantity; however, the feed quality is compromised in large-scale farms that offer excess amounts of grains to make up for the limited availability of the greens [16]. Such compromised feeding practices, which may not provide the required proportions of satiating components such as roughage and concentrates [51], can result in prolonged hunger. Moreover, the animal’s hunger seems to be exploited by the industry to ensure minimum resistance to milking: the majority of the farms in the leading dairy states of Uttar Pradesh and Rajasthan provide the bulk of the total feed just before milking the animal [52,53]. Access to water, a nutrient naturally consumed 7 to 12 times a day by bovines irrespective of their age [54], is generally provided for short intervals and only twice or thrice daily [53,55,56,57], a state far from the desired ideal of ad libitum water availability. Ponds and groundwater are common sources of water [53,55], and are prone to being contaminated by chemical and biological pollutants [58,59], reducing the quality of the water consumed.

#### 3.2.2. Bull Breeding Centers

The general inaccessibility to dietary protein and energy sources in the nation is considered as an influential factor on the breeding bulls’ diet [60]; however, the actual feed and water availability to the bulls remains fairly unexplored. No up-to-date national guidelines are available for the breeding-focused dietary requirements of neither cow bulls [61] nor buffalo bulls [62], which can risk towards a compromised dietary status of the bulls. Further on-field studies are warranted to understand the dietary status of these bulls.

#### 3.2.3. Animal Shelters

The majority of the uneconomical bovines that land in a cow shelter are deprived of the freedom from hunger and thirst, with more than two-thirds of the shelters offering inadequate feed and water [17]. The inadequacy is reported as high as 80% in the government-run shelters [17]. The staff’s limited awareness of science-based welfare practices [63], like the provision of adequate and appropriate feed [64], can further hamper wellbeing. The total amount of milk provided for the calves is questionable in shelters given their dairy farm-like milking practices that separate the calf from the mother shortly after birth and permit suckling only briefly before milking [17]. In the shelters run by religious institutes, food offered by the frequent visitors [65], such as non-centrifugal cane sugar and Indian flat bread [66,67], do not form a part of the animal’s natural diet and can impact the animals’ diet quality. For instance, sugar increases feed intake in bovines [68], which in a situation of compromised feed availability will draw the animal further away from the freedom of hunger. Unsaturated cooking oils used in the flat breads decrease the natural fermentation in bovine stomach and reduce digestibility [69]. Sub-optimal management practices such as inadequate cleaning of the feed and water troughs and lack of washing the enclosures [17], potentially due to the commonly observed lack of resources [17] degrade the quality of the feed and water consumed by the animals.

#### 3.2.4. On the Streets

The nation has over five million bovines on the streets [70], who seek feed and water from unideal and inadequate resources such as dustbins, landfills, crops and local water bodies [71]. Bovines which are not abandoned but are let out on the streets routinely by the farms or shelters to fend food [72] (p. 130) are often physically restrained through roping that ties their foot to their neck or to the foot of another animal [40], reducing the animal’s ability to seek food or water. The plight of finding food is exacerbated in developing, pregnant, and lactating animals given the limited feed resources available to satiate the body’s increased nutritional demand [73,74]. Calves that are naturally drawn to bovine milk as their primary feed source [75] may especially find it hard to seek and digest food off the streets. A lowered scavenging ability resulting from physiological causes such as injuries, diseases, and old age further reduces the likelihood of being able to obtain food and water. Attempts at feeding off human food resources and/or food waste come with considerable welfare costs such as acid attacks by farmers for invading crop fields [76,77,78], competition with scavenging omnivores like dogs, crows, and monkeys [79,80] which are capable of attacking the bovines, and ingestion of an array of inedible waste such as plastic and sharp solids including nails, needles, glass, and wires [81,82]. Such pain-inducing consequences exhibit the potential to reinforce the avoidance of food scavenging in those areas and/or limit the animal’s physical ability to scavenge, thereby only worsening their hunger and thirst.

#### 3.2.5. Transportation

Regardless of the existence of a national law which provides clear guidelines for feeding and watering animals in transit [83], the bovines’ freedom from hunger and thirst is subject to the legality of the transportation and the availability of resources. Transportation of bovines occurs both legally and illegally [18,84]. Illegal road transportation of bovines to reach places that permit their slaughter is typically long-distance and devoid of any breaks for the animals [72] (p. 222), thereby diminishing the animals’ access to feed and water for prolonged durations. Legal road transportation too, however, does not guarantee access to adequate feed and water [29,85], with bovines being diagnosed as famished and dehydrated upon arrival in markets [29]. Animals transported on foot across national boundaries for slaughter are meagerly fed and watered until they reach the abattoir [72] (p. 222). The feeding regime of bovines transported on foot to markets remains largely unstudied; however, the access of such animals to the frequent roadside greens in the non-desert areas [86] is possible.

#### 3.2.6. Animal Markets and Slaughterhouses

The relatively brief period the animals spend in markets or an abattoir lairage upon completing a transport, are often laden with continued hunger and thirst. Unavailability of feed and water resources in markets [72,87] (p. 234), accompanied by malpractices such as feeding thirst-inducing herbs to mask physical deformities that can reduce the animal’s market value [88], deprive the animal of satiation. Although pre-slaughter withholding of feed is a recommended abattoir practice to ease the slaughter process [89], Indian bovines are additionally deprived the access to potable water before slaughter [72] (p. 234, p. 274), thereby aggravating their starvation.

### 3.3. Freedom from Discomfort

#### 3.3.1. Dairy Farms

The prevalence of sub-standard infrastructure in the small-scale dairy farms indicates the absence of an appropriate physical environment including shelter and a comfortable resting area for the animals. Makeshift shelter arrangements such as tin sheds, open road-side stalls, and road-side tarpaulin sheds are common in the small-scale dairies [16,90], and can result in lowered hygiene, lack of ventilation, and minimal protection from stress-causing events such as exposure to inclement weather, vehicle and pedestrian traffic, and stray animals [91,92]. Space limitations in these farms, which can stem from the need to rear unproductive animals, often result in overcrowding of the animals, the transitional use of the housing area as the milking parlor and in a failure to quarantine sick animals [16,40], decreasing the suitability of the farm as an ideal environment for the animals. Housing animals on concrete, non-grooved floors is typical [93], with the majority of the bovines lacking any access to soft ground [40]. Such slippery flooring has been associated with hoof, knee, and hock injuries in the bovines [94,95,96]. Bone damage under such circumstances is likely, however, the documentation of bone injuries in the literature seems limited due to the associated pre-timely disposal of such economically burdensome animals [97]. The provision of a floor bedding is not a prevalent practice [16,40,98], which coupled with the undesirable temperatures of the high thermal conductive concrete [99], diminishes the animals’ ability to rest. The moisture-absorbing concrete [100] along with the animal’s excreta, can create a breeding ground for microbial growth and promotes infectious diseases of the hooves and udder, and a lower milk production in the bovines [94,101,102,103]. Nearly a third of the dairies lack adequate lighting in the animal housing [40], reducing the bovine’s ability to visually navigate their environment.

The prevalence of non-ideal housing arrangements such as hard flooring, inferior roofing materials, and low ceiling height has been observed in the large-scale dairies only sporadically [16], implying the higher suitability of such organized farms in providing the animals a desirable environment.

#### 3.3.2. Bull Breeding Centers

Bulls reared in breeding centers are exposed to a more customized physical environment than the average small-scale dairy farm female; however, the appropriateness of the environment to satisfy the freedom from discomfort remains questionable. Similar to dairy farms, concrete flooring is a common characteristic of the bulls’ housing [104,105], depriving the bulls of access to soft ground. Although the bovine breeding guidelines require a loafing area to be a part of the housing [106], several centers fail to provide it [16,104]. Centers housing the bulls in individual pens with concrete walls along the length of a rectangular pen [16] significantly impact the animal’s ability to move, view his surroundings, and interact with his conspecifics. Centers with limited space practice group housing and continuous tethering of the bulls [107], which reduces the resting space available per animal and inhibits their movement. The welfare compatibility of the exercise and semen collection environments in breeding centers remains largely unknown and demands more primary research in the area.

#### 3.3.3. Animal Shelters

The dependency of the shelters on external funds for infrastructure and operations plays a determining role in the animals’ freedom from discomfort. A majority of the shelters are dependent on donor and government funds for infrastructure development, run in limited, brick-flooring spaces, and consist of technically untrained or minimally trained staff [17,64,108]. Housing of shelter animals in makeshift spaces including in the vicinity of temples and abandoned parking lots has also been recorded [72] (p. 144). Infrastructure as such, which is inadequate and/or not specifically designed for dairy bovines, can minimize the animal’s access to soft ground, the capacity to rest, and similar to the small-scale dairy farms, can increase exposure to pathogens and natural and man-made stressors [91,92]. Moreover, such arrangements make the animals susceptible to injuries: inappropriate shed flooring has resulted in increased lameness occurrences [109] and the sharp ends of the shelter housing have caused skin injuries in the bovines [110]. The staff’s lack of awareness which leads to the use of scientifically inappropriate management practices such as continuous tethering with short (<1 m length) ropes [17,64], minimal sanitization of the enclosures [17,108] and limited biosecurity measures [110], further reduces the bovines’ access to an appropriate environment.

#### 3.3.4. On the Streets

Abandoned street bovines are chronically deprived of an appropriate environment. In the meagre availability of a shelter [17], several street bovines lack access to a constant, comfortable resting area and are exposed to multiple stress-inducing environmental events that commonly affect the small-scale dairy farm animals [91,92]. The need to scavenge for food causes constant physical displacement and can subject the bovines to multiple undesirable, life-threatening environments which are either severely compromised on hygiene (e.g., dustbins and landfills), prone to attacks by humans (e.g., during an animal’s intrusion in crop fields and in human habitats), and/or prone to traffic accidents (e.g., on roads and rail tracks) [76,77,78,111,112,113,114].

#### 3.3.5. Transportation

The bovines’ transportation environment tends to be subpar to the recommended government standards [83], with the legal transportation providing an environment only minimally better than an illegal one. The farm’s financial capacity plays a crucial role in the access to appropriate transportation techniques, making large-scale farms capable of utilizing more welfare-friendly techniques than the commonplace small-scale ones. For instance, large-scale dairies utilize a plank to load the animals unlike the small-scale ones which utilize more makeshift approaches such as loading using a heap of soil [115]. Animal handling assistance during loading is more common in the larger farms, with the small-scale ones having minimal to no handling assistance [115], possibly due to the limited financial capacity to hire manpower in the latter. Transportation in makeshift carriers, such as goods carriers or closed milk transport tanks, is common regardless of the dairy size and the legality of the transportation [18,84]. Carriers not designed for animal transportation can be a potential source of discomfort given the inappropriate vehicular design. For instance, a commonly bovine transportation vehicle, the Tata Ace open-air mini truck, with side walls merely 12 inches high [72,115], cannot support the animal in balancing their body without the risk of falling off the truck. Inappropriate flooring of such vehicles coupled with the lack of bedding [115] lessen the animal’s comfort further. Closed milk transport tanks, often used for illegal transportation [72] (p. 216), create undesirable environmental conditions by exposing the bovines to excreta, blood, and to sick or dead animals for extended time periods due to the lack of stops during the journey. Overloading and continuous tethering during transportation is typical in both legal and illegal bovine transportation [18,84,115], which minimizes the space available per animal, hampers ventilation, and decreases the environmental sanitation levels.

#### 3.3.6. Animal Markets

Cattle markets lack a concrete unloading and loading arrangement [87,116], leading to the use of unideal loading and unloading techniques such as the use of manual coercion [18]. Shelter areas are not a common feature of cattle markets [87,116] which coupled with inappropriate tethering practices such as tethering multiple animals to a single anchor [72] (p. 233), can severely impact the animal’s access to a resting area. Markets often lack an organized waste management system [87], which can degrade environmental hygiene.

#### 3.3.7. Slaughterhouses

The lack of an appropriate environment just prior to and during slaughter becomes concerning for several bovines, especially cows who are slaughtered illegally [72] (p. 251). Illegal and/or small-scale slaughterhouses located in villages and city slums [72] (p. 252), tend to lack basic amenities such as a resting area, water, electricity, and appropriate equipment [117] which can result in discomfort before and during the slaughter. Resting areas are more frequent in the legal and thereby more organized slaughterhouses [72,118]; however, similar to the small-scale abattoirs, the larger ones are not necessarily free of microbial contamination [118,119], resulting in a compromised hygiene of the surroundings including the lairage.

### 3.4. Freedom from Pain, Injury, and Disease

While the direct measure of an animal’s pain perception might not be feasible, pain is closely associated with the presence of injuries and diseases. The following section analyzes the factors influencing the occurrence and characteristics of the injuries and diseases afflicting the Indian dairy bovines.

#### 3.4.1. Dairy Farms

The farmer’s lack of awareness about and/or limited financial affordability of veterinary care appear as the driving factors of the health status of bovines in the small-scale farms. Despite the availability of free or nominally charged government-provided veterinary services [120,121], more than four-fifth of the small-scale dairy farms provide veterinary care only when the animal appears sick and shows a reduction in milk production [40]. The situation is aggravated by the significant 49% gap between the required and available on-field clinical veterinarians [122], which results in farmers opting for the services of the modestly equipped on-field para-veterinarians, given the farmer’s inability to transport the animal to the government clinic [121,123,124]. Such limited health care focus, can result in a lack of preventive veterinary care provision such as nutritional planning, deworming and vaccinations, quarantining of new animals, scientific breeding and pregnancy management, calving and milking management. Limited health care focus can also curtail curative veterinary care provision such as diagnosing and treating of wounds and ailments that may not exhibit immediate effects on the animal’s milk production, and thereby exert significant impacts on the animal’s physical health.

Multiple outbreaks of infectious bacterial diseases such as Johne’s disease [125,126,127], fungal diseases such as aflatoxicosis [128,129], and viral diseases including Buffalopox [130,131,132] and Brucellosis [133,134,135] have been reported consistently in Indian bovines. Despite being preventable through proper vaccination, many infectious diseases such as foot and mouth disease, blue tongue, infectious bovine rhinotracheitis, and hemorrhagic septicemia are endemic to India and therefore common in the bovines [126]. The repetitive occurrence of reproductive disorders, including pyometra, metritis, dystocia, vaginal and cervical prolapse, and post-partum retention of the fetal membranes is common in the bovines [136,137,138,139,140], and serves as a leading cause of buffalo slaughter [141] and potentially cow abandonment. Lower breeding success is rampant in farms that utilize para-veterinary services [121,124] and in farms that rear the silent heat-exhibiting bovine, the buffalo [142,143,144], leading to repetitive breeding attempts which can inflict injuries to the reproductive tract and make the animal prone to infections such as pyometra. The repetitive breeding syndrome, characterized by the animal’s inability to conceive even after multiple breeding attempts, afflicts as many as 40% of the dairy bovines in small-scale farms [40,145]. Lactational metabolic ailments are not uncommon in the farms and include mastitis [146,147,148,149], milk fever [150,151,152], ketosis [150,153,154,155], and downer cow syndrome [156,157]. Such metabolic diseases which can reduce the animal’s milk production, create a vicious economic loop characterized by a higher monetary cost of rearing the sick bovine [158,159], resulting in lower affordability of veterinary services, which can lead to the lingering of the animal’s ill-health, and in turn reduce the profits from the sale of their milk.

Prevalent milking malpractices such as the abuse of the hormonal drug Oxytocin to induce milk let down [40,160], blowing air or thrusting tail into the vagina to simulate milk let down (commonly termed phooka or doom dev, a practice prohibited by law) [161], and milking of ill bovines [40] notably increase the animals’ susceptibility to pain, injury, and disease. Other pain-inducing malpractices include one-time procedures such as unanesthetized de-horning and de-budding [40] and repetitive, regular procedures such as coercive handling by pulling the nose rope or twisting the tail [29].

A relatively small number of dairy farms, that largely comprise the limited cluster of the large-scale farms, tend to have assigned in-house veterinarians [16], and/or are run by relatively educated farmers with adequate awareness [44,108,162], thereby maintaining and promoting the productive animals’ physical health. Measures taken by such farms towards the veterinary care of the financially burdensome bovines, if any, are under-represented in the literature and warrant more primary research.

#### 3.4.2. Bull Breeding Centers

Studies analyzing the identification and treatment of injuries and ailments in breeding bulls are scarce. However, the modest housing conditions of the bulls including the use of concrete flooring in the pens and the absence of a loafing area [16,104,105] can make them prone to physical environment-related injuries such as head wounds through stereotypical head banging on the enclosure walls [16] and social environment-related injuries such as those arising from agonistic conspecific interactions in group-housed bulls [162]. Understanding the presence of breeding-related body damage in these bulls, for instance injuries and diseases arising from the semen collection process, and the measures taken for its diagnosis and treatment, demands further primary investigations.

#### 3.4.3. Animal Shelters

Akin to the small-scale dairy farms, an inadequacy of veterinary care accessibility is a characteristic of the majority of animal shelters. Several shelters lack a consistent revenue model [17] which can hinder the access to quality veterinary services. Less than a quarter of the shelters have resident veterinarians, with the bovines in the remaining 80% of shelters lacking regular access to veterinary services [17]. Inadequate staff training exacerbates this problem by reducing appropriate injury and disease surveillance and treatment protocols such as animal quarantining [17,163]. This absence of appropriate veterinary and management practices, accompanied by the general prohibition of slaughtering cows, can result in chronic pain and prolonged suffering for the injured and/or ill animals. A variety of injuries have been reported in the shelter bovines and include skin injuries such as lesions, swellings, teat and udder injuries, and lameness-associated trauma such as hock and carpal joint injuries and hoof and claw overgrowth [64,108,110]. Shelters commonly house less productive or unproductive ex-dairy bovines [164], who, given the significant prevalence of clinical and sub-clinical diseases in the dairy farms (Refer to the Dairy farms sub-section of the Freedom from pain, injury, and disease section of this article), may not always be free from ailments. Reproductive diseases are common in the shelter bovines [64,108,165] and if transmissible, can impact the herd health, as seen in the cases of Brucellosis occurrences in multiple bovine shelters [166,167].

#### 3.4.4. On the Streets

Bovines are exposed to a variety of pain-inducing stimuli on the streets such as through the ingestion of inedible waste, attacks by humans and other scavengers, and anthropogenic accidents [76,77,78,79,80,81,82], which can result in injuries, post-traumatic deformities, disorders, and can increase the animal’s susceptibility to diseases by debilitating their immune system. For instance, ruminal impaction, resulting from the ingestion of inedible waste has been commonly reported to culminate in mortality in the stray bovines [168]. The stray bovines carry some transmissible diseases that afflict the dairy cows such as Brucellosis [169]; however, understanding the array, severity, and intra- and inter-species spread of these diseases calls for further primary research. With veterinary care availability limited to that provided by the meagre bovine-oriented or bovine-care-inclusive non-profit organizations [170], injured or ill street-dwelling bovines can be subject to chronic physical suffering until they succumb to death.

#### 3.4.5. Transportation

Limited access to dedicated bovine transportation facilities [18,84,115] and [72] (p. 216) directly influences the animal’s welfare around and during the transport. Pain-inducing handling practices such as twisting the septum-tied nose rope, tail twisting, prodding with sticks and stuffing eyes with chilies to induce movement in and out of the vehicle causes nose, eye, tail and skin injuries, fractures and oculo-nasal discharge [18,28,29,30,31,115]. Commonly observed bodily damages at the end of a transportation include bruises, hyperkeratosis, fractures, and oculo-nasal discharge [18,28,115,171], which are reflective of the handling, are only worsened by the inappropriate transportation environment. Although the legal transportation of cows has shown to result in injuries and deaths [72] (p. 235) and [84], illegal transport, which occurs in concealed vehicles and during the darker hours [18], can increase the occurrence of disease transmission, injuries, and accidental deaths multi-fold.

#### 3.4.6. Animal Markets

Injuries and diseases in the Indian bovine markets appear to be under-researched; however, the environmental conditions and the animal handling practices in the markets possess a potential to cause physical harm to the animal. Disease and injury may result from the lack of systematic organization of the animal markets: the occurrence of wounds and the spread of diseases through the lack of animal health disclosure requirements, unavailability of in-market veterinary services, inappropriate waste management, and the absence of adequate unloading, resting and loading facilities [87,116] merits further research. Pain-inducing handling practices in the markets, such as stuffing eyes with itchy substances like ginger, lemon juice, and splashing water in the nostrils to induce the fear of drowning [72] (p. 230) can directly cause bodily damage.

#### 3.4.7. Slaughterhouses

Pain- and injury-inducing stimuli which prolong the animal’s suffering before death appear rampant in the bovine slaughterhouses, not due to the act of slaughter itself, but rather due to the inappropriate handling methods and the sparse environmental hygiene of the slaughterhouse. Inadequate resources and amenities in the small-scale slaughterhouses [117] as well as in some large-scale abattoirs [118] promote pain and injury-causing practices such as handling the animal by their ears and using vaginal or anal electric prods, lack of stunning or repetitive stunning attempts with blunt force, and cutting the legs while the animal is still conscious, the latter in an attempt to avoid injuries to the workers [72] (pp. 256–275). Numerous disease-causing agents such as *Listeria* spp., Staphylococcus aureus, Escherichia coli, Schistosoma nasale, *Leptospira* spp., and *Pasteurella* spp. [172,173,174,175,176] have been recorded in the bovines and their beef in levels high enough to cause the disease; however, it is not conclusive if the animals contract the diseases in the slaughterhouse or earlier.

### 3.5. Freedom to Express Normal Behavior

Unlike the other four freedoms which focus on the passive removal of barriers to provide comfort (“freedoms from”), the “freedom to” express normal behavior provides the animal with an ability to control the actions they can perform to attain comfort, making the latter a crucial measure of the animal’s wellbeing. However, the definition of what constitutes normal behavior may vary from one management environment to another. Therefore, the below-performed analysis of this freedom in the Indian dairy bovines focuses on the attainment or the lack of the commonly acknowledged normal behaviors seen in domestic cattle, viz. social behaviors such as maternal, sexual and grooming behaviors, as well as maintenance behaviors such as feeding, walking, and resting [177].

#### 3.5.1. Dairy Farms

Calf-rearing practices, in both large- and small-scale farms, reduce the animals’ ability to exhibit normal behaviors. The Indian day-old weaning practice which allows for calf–mother contact only briefly before milking [44,178] hampers the calf’s frequency and duration of expressing normal behaviors towards the mother [179]. Suckling, a behavior that is generally performed every two to three hours by a calf [180], is restricted to only two short occasions in a day [16]. Other social behaviors such as sniffing, licking, and rubbing against the mother [179] stand meagre chances of occurrence in the brief calf–mother interactions which are focused on the act of suckling. Calves born in large-scale farms or those born to the only bovine of a small-scale farm are reared in isolation until the attainment of puberty [16], depriving them from exposure to animals of either sex, and diminishing the long-term ability and willingness to perform social behaviors with conspecifics as seen through an avoidance of other animals by these calves [181,182]. Isolation can further lead to coping behaviors such as vocalization, pacing, and abnormal oral activity [183,184,185], which reduces the time the calves can spend performing normal behaviors.

Adult bovines are unable to express a variety of social and maintenance behaviors on the dairy farms. Although the prevalence of artificial insemination as an impregnation technique in dairy cattle has increased worldwide [186,187], its impacts on the animal’s ability to perform normal sexual behaviors cannot be ignored while analyzing the Indian dairy bovines’ welfare. Commonplace improper and repetitive artificial insemination attempts [121,124] can be painful, stress-inducing, and can create a negative association with sexual behaviors, thereby diminishing the expression of courtship and mating behaviors. Expression of pre-partum behaviors such as restless movements and preference for isolation [177] may not be feasible given the inadequate space availability per animal [16,40,90]. The common occurrence of dystocia cases, many of which are prolonged and do not attain a scientific resolution [37,38,39], hinders the animal’s ability to perform the normal act of parturition. Early weaning practices [44,178] drastically reduce the frequency and duration of maternal behaviors such as nursing, licking, and grooming the calf [177]. The use of khalbaccha (an effigy made with stuffing a dead calf’s body with hay to mimic a calf and induce milk let down), commonly used for dairy buffaloes [40], reduces the animal’s control on initiating the nursing behavior and reduces the performance of concurrent maternal behaviors, making the normalcy of this induced behavior questionable. Coping acts such as vocalization, inactivity, and reduced calf acceptance by the mother, which stem from an inability to perform normal behaviors [182,188], further reduce the time the animal is able to spend performing normal behaviors. Inadequate management practices including the lack of an appropriate physical shelter and a comfortable resting area [16,40,90,93,98], compromised feeding practices [16,52,53] and restricted water access [53,55,56,57] limit the animal’s ability to perform maintenance behaviors such as resting, walking, rubbing against objects, scratching themselves with limbs, licking themselves, rubbing their neck on the ground, grazing, browsing, ruminating, and drinking.

#### 3.5.2. Bull Breeding Centers

The breeding bulls’ ability to express normal social and maintenance behaviors is curbed through a variety of management practices. The standard semen collection process that involves the use of a dummy mate and a semen collector [189,190] induces the normal act of mounting and provides a simulated environment for ejaculation behavior; however, the process does not permit the expression of normal courtship behaviors. Confinement and the continuous tethering of the bulls [16,107] blocks the opportunity to perform maintenance behaviors of walking, grazing, self-grooming, and reduces the ability to perform social behaviors such as allogrooming [179]. The coping behaviors observed in these bulls including head banging [16] and agonistic conspecific interactions [162], further reduce the time spent performing normal behaviors.

#### 3.5.3. Animal Shelters

Factors impacting the expression of normal behaviors in animal shelters bovines are analogous to those affecting the dairy farm animals, and can be further compounded by a lack of funds. The common practice of repetitively inseminating the post-productive animals in an attempt to obtain economic gains from milk production [17] curtails the animal’s ability to express normal courtship and mating behaviors. Exposure to frequently changing social environments due to exposure to unknown human visitors in religious shelters, calf–mother separation and brief interactions just prior to milking, or the arbitrary addition of new bovines [17,65], can drastically reduce the animals’ ability to express normal social behaviors including maternal and allogrooming behaviors. Inadequate feed availability and/or inappropriate feeding by staff and visitors [17,64,65] can reduce normal maintenance behaviors such as grazing, browsing, rumination and drinking, while the limited space availability due to makeshift arrangements and/or continuous tethering practices [72] (p. 144) and [17,191] can impact the animals’ expression of resting, sleeping, walking, and self-grooming behaviors. Chronic injuries and diseases which can go untreated given the scarcity of veterinary care [17,163] can significantly reduce the animals’ expression of normal behaviors.

#### 3.5.4. On the Streets

The street bovines, especially the abandoned ones, have the freedom to express a larger variety of normal behaviors as compared to their dairy and shelter counterparts given the former’s lack of ownership by humans. Social behaviors such as sniffing, standing beside and walking with other animals [179] are feasible for street bovines in the presence of two or more amicable street bovines. Maintenance behaviors like walking, sitting, scratching themselves with limbs, licking their own body, rubbing against objects and browsing [179] can be performed by these bovines but are subject to unpredictable human-related interruptions, such as attack by humans (e.g., during an animal’s intrusion in crop fields and in human habitats) and traffic accidents (e.g., on roads and rail tracks) [76,77,78,113,114]. The quantity and the quality of the feed and water the bovines consume is often lower than desirable [71,79,80,82] and can reduce the animal’s ability to perform maintenance behaviors of ruminating and drinking. Chronic physical suffering through wounds and illness including acid attacks by humans, ruminal impaction due to ingestion of inedible waste, and contracting transmissible diseases like Brucellosis [76,77,78,168,169] can prohibit the animal from being able to perform the normal behavioral repertoire.

#### 3.5.5. Transportation

Worldwide, transportation environments of livestock are not ideal for the display of the majority of social and maintenance behaviors; however, this condition is worsened in India, given the inferior quality of bovine transportation characterized by makeshift transportation arrangements and/or illegal transportation. The lack of stops during illegal long-distance transports [72] (p. 222) and overcrowding and continuous tethering in both legal and illegal transports [18,84,115] inhibit the performance of feed ingestion, rumination and drinking, and can reduce control over performing the normal act of urination and defecation. Such arrangements can furthermore decrease the animal’s ability to sit and rest and can diminish the expression of social behaviors such as suckling, licking, and allogrooming.

#### 3.5.6. Animal Markets and Slaughterhouses

Both bovine markets and slaughterhouses expose the animals to undesirable, novel physical and social environments known to induce stress and fear [18,117,118,192,193], thereby reducing the performance of normal behaviors. Ingestion and drinking behaviors are reduced owing to the substandard and harmful feeding and watering practices [87,88] and [73] (p. 234), while the frequent lack of shelters [87,116,117] can impact sitting and resting behaviors. Overcrowding in markets [72] (p. 233) and exposure to the conspecific blood and urine in slaughterhouses [117,118] can curb social behaviors such as allogrooming, suckling, and nursing.

### 3.6. Freedom from Fear and Distress

Occurring in multiple phases of the Indian bovines’ lives, fear and distress is not a stand-alone phenomenon but is intricately linked with and can be aggravated by the compromised state of the other four freedoms of welfare.

#### 3.6.1. Dairy Farms

Peri-natal calf management practices in India, such as the unscientific resolution of prolonged dystocia [37,38,39], day-old weaning [44,178], and the neglect of the calves lacking utility [40] makes animals prone to fear and distress by creating physiological and physical stress-inducing circumstances. Although unexplored in the Indian context, studies on dairy bovines worldwide indicate the presence of physical stress markers (e.g., reduced colostrum consumption, reduced vitality) and physiological stress markers (e.g., high cortisol levels, lowered passive immunity) in calves surviving prolonged dystocia [194,195,196,197], meriting research on the effects of prolonged dystocia in Indian dairy calves. Day-old weaning of Indian calves has been shown to cause abnormal oral behaviors in calves, such as floor licking, which are associated with fear-coping mechanisms [178,198]. The lack of consistent contact with the mother and limited suckling opportunities have resulted in physiological irregularities such as lowered Ig immunoglobulins, higher cortisol levels, and the occurrence of diarrhea in the Indian calves [45,199,200,201,202,203], indicating the presence of fear and distress in these animals.

Exposure to acute and chronic afflictions such as recurrent diseases [125,126,127,128,129,130,131,132,133,134,135], and injuries resulting from inappropriate housing infrastructure [94,95,96,101,102] make the bovines prone to pain-induced fear [204]. Inappropriate reproduction-related practices, such as improper and repetitive artificial insemination attempts [121,124], frequent and prolonged dystocia [37,38,39], the practice of phooka [160], and bestiality [95,205,206] can further intensify the animal’s fear. Fear-indicating behavioral expressions such as avoidance of the stockperson, restlessness and resistance during milking, agonistic behaviors towards conspecifics and stereotypic behaviors such as head shaking have been observed in these animals [52,207,208,209].

#### 3.6.2. Bull Breeding Centers

Little is known about the prevalence of fear and distress in the breeding bulls. Studies reporting stereotypies and agonistic interactions with conspecifics [16,162] suggest fear and stress-induced behavioral changes. Such behavior, if pain-inducing, can result in the consequent fear of pain in the bulls. However, the expanse and frequency of fear and stress-related changes in these animals warrants further research.

#### 3.6.3. Animal Shelters

For animals in shelters, frequent occurrence of injuries [64,110,210] and reproductive diseases [108,165,166,191,210], accompanied by limited veterinary care of the bovines [17,163], can lead to pain-induced and/or chronic stress, as reflected through high hair cortisol levels observed in these animals [211]. A history of fear and/or pain-inducing experiences with humans in the dairy farms can intensify the fear of humans in these ex-dairy animals as observed in some shelters in the southern parts of the nation [212]. However, the absence of the fear of humans has been reported in other shelter bovines [109,213], suggesting the need of further research around human-related fear in shelter bovines. Considering the ever-changing social environment of the shelters, where new bovines are arbitrarily added to the shelters [17], studying the amiability of the animals’ behaviors with their conspecifics may provide additional insight into the prevalence of fear and distress in the bovines.

#### 3.6.4. On the Streets

A direct measurement of fear and distress in the street bovines can be logistically difficult given the fluidity of their resting and feeding areas, coupled with a communal outlook rather than an individual ownership approach towards these animals. The bovines are exposed to a variety of anthropogenic pain-inducing stimuli such as attacks by humans and traffic accidents [76,77,78,113,114] which can result in fear of the perception of potential pain associated with the stimulus [204]. Fear in prey species like cattle can also arise in the vicinity of predator species [214,215], including dogs and crows with whom the street bovines are known to share scavenging spaces [79,80]. Irregularity in feeding success, which is typical for street bovines [71], is frequently associated with the display of stereotypic behaviors [216,217,218], an indicator of stress.

#### 3.6.5. Transportation

Commonly deployed sub-par transportation practices such as long transportation periods devoid of stops, compromised space and ventilation conditions, the pulling of the animal with a tractor, and a lack of gender-based segregation during transportation [18,84,115] are shown to result in the exhibition of stress-related behaviors including frequent urination and defecation, resistance and exhaustion [85]. Furthermore, the lack of adequate feed and water access [29,85], limited possibility of physical movement and being surrounded by known and unknown animals [18,84,115] exhibit the potential to intensify the animals’ fear and distress.

#### 3.6.6. Animal Markets

The bovine market is a novel place for several of the animals, leading to a natural apprehension of the market area [193] and a consequent enhancement of the fear induced by exposure to inappropriate management practices in the market. Management practices such as feeding malpractices [87,88] and [72] (p. 234), unethical animal handling practices [18] and [72] (p. 230), and lack of adequate shelter [87,116] can cause stress by depriving the animal of their basic maintenance requirements and/or by inducing pain. However, studies measuring fear or stress of the animals in the markets are lacking and point to an area that will merit research.

#### 3.6.7. Slaughterhouses

Similar to the markets, slaughterhouses expose the animals to novel physical and social surroundings; however, the environment in the slaughterhouses can be more mentally impactful given its pain-causing ability and the variety of fear-inducing sounds, smells and sights it exposes the animals to. The common makeshift practices, such as handling by ears, use of vaginal or anal electric prods, and repetitive blunt force stunning attempts [117,118] and [72] (pp. 256–275) which can cause pain and prolong the slaughter duration, can promote fear and helplessness-induced stress in the bovines. Noise exposure in the slaughterhouse, including human sounds during animal handling, and metal clanging sounds such as those of metal gates and metallic equipment are known to induce fear in cattle as seen through increased movements and heart rate [192]. The smell of conspecific blood and urine from stressed conspecifics can further intensify fear in cattle as expressed through increased movements [219]. The inability to fight or flight resulting from the often compromised health status of the slaughter animal alongside the lack of adequate resources and amenities in the slaughterhouses [117] can heighten fear and stress in these animals.

## 4. Discussion

The Indian dairy bovines are deprived of one or more of the Five Freedoms of animal welfare in multiple phases of their lives, signifying an overall compromised quality of life in these animals. The prevalence of hunger and thirst is common in the bovines, followed by the presence of physical discomfort, and pain, injury and disease. The freedom to express normal behavior is met partly across life phases, plausibly due to the wide variety of behaviors that constitute the category. The presence of fear and distress is often visible as a ramification of the compromised status of the other four freedoms in these animals, and its intensity varies from one life phase to another.

During the juvenile and productive phases of the bovines, the attainment of the freedoms appears to be directly dependent on the size and financial capacity of the rearing facility. The minority-numbered large-scale, organized dairy farms are well-resourced to deploy management practices [16,40,57,120,220] that can lead to at least a partial provision of most of the freedoms, unlike the ubiquitous small-scale farms which cannot afford the welfare-friendly practices [16,40,90,98]. The purpose of the animal in the dairy industry, which governs the type of facility they are reared in, can also influence the achievement of the freedoms. For example, bulls in breeding centers are exposed to a more organized environment [16,104,105,107] that can promote better management practices, unlike those available to the females reared in the typical small-scale dairy farms. However, studies measuring the breeding center bulls’ welfare are meager, warranting further research to holistically understand the welfare impacts of rearing bulls in such centers.

A parallel loss of two or more of the freedoms emerges as a common theme in the post-productive lives of the bovines, reinstating the subdued yet crucial role of the animal’s utility value in the rearing of these bovines. A majority of freedoms are simultaneously compromised in phases which mostly occur after the farmer has culminated the animal’s ownership: on the street, during transportation, in the animal markets, and in the slaughterhouse. For the shelter animals, the purpose of the animal shelter determines the achievement of the freedoms, with the animal rescue-oriented shelters practicing more welfare friendly management than the government run shelters, and the shelters run by religious institutes [17,63,108,191,210]. However, a lack of adequate financial funding comes across as a common problem in the shelters and is closely related to the lack of attainment of more than one freedom.

These results exhibit a failure to appropriately interpret and implement the legislation surrounding bovine husbandry. The current management approach is heavily dependent on the species of the bovine reared and the finances available to the rearer, which often leads to disregarding the utilization of modern and scientific practices as recommended in Article 48. The variations in the implementation of Article 48-based laws across states impact the welfare of the bovines by affecting the post-production lives of cows and the distance they travel, legally or illegally, to be slaughtered. A recent rise in the abuse of the slaughter ban regulations as a cow vigilante political tool for “protecting” the cow under religious pretext [212,221] displays an objectification of the animal, and demands inquiry to understand the welfare implications of such legally justified de-animalization of the cows.

The results shed light on the ironic effects of the current Hindu religious faith on India’s bovine husbandry. By influencing the bovine-related legislation and the social norms of the nation, these beliefs have directly influenced the welfare of the bovines. The cow slaughter ban arising from the religious beliefs that idolize the cow ultimately causes the post-production abandonment of the animals on streets or in shelters, both being places where the animals’ welfare is compromised. The absence of legal regulations for buffalo husbandry given a thorough disregard of the animal’s religious significance makes the buffaloes a dispensable commodity whose welfare is not a primary concern for their rearer.

This compromised welfare state of the Indian dairy bovines manifests itself as a multifactorial outcome of the socio-religious, economic and legal influences around the bovine (Figure 4). Across nations, dairy animal welfare is influenced by the rearer’s financial ability [222,223,224,225,226,227,228] and attitude towards the animals’ welfare [228,229,230,231,232,233]. In the Indian scenario, the two welfare-determining factors of financial ability and attitude, appear to be regulated by the extent of the legal binding around the bovine species being reared. The presence of laws preventing the slaughter of cows and the absence of those around that of buffaloes, both largely a product of the nation’s religious article of faith, can determine the rearer’s attitude towards rearing the animal by shaping the societal norms around that bovine species. These slaughter-banning laws can further determine the animal’s welfare by impacting the rearing facility’s economics: the inability to slaughter animals can result in losses through the inability to sell non-productive or sub-productive animals and the obligation to rear them until their natural death.

This compound, culturally defined, species-specific dynamic of dairy bovine welfare in India presents itself as a paradoxical situation where the acclaimed sacrality of the cow and the slaughter ban arising from this contemporary faith, shapes the welfare of the cows negatively, and nurtures a complete disregard for the welfare of the buffaloes (The Cow Paradox). The Cow Paradox challenges the assumptions around the welfare status of Indian dairy bovines, by questioning the purpose (and therefore the effects) of the legalities which ban the slaughter of certain bovines while permitting that of the rest. It paves a way to better analyze the welfare of Indian bovines by reviewing it through the socio-cultural lens of human–bovine interactions and relationships. This hypothesis calls for further research to understand the role of the bovine-related laws in shaping the sale, purchase and/or abandonment of the bovines, in influencing the management practices surrounding the bovines, and in determining the finances of the dairy industry at the local, state and national levels, in order to obtain a deeper appreciation of the effects of legislative influences on the bovines’ life quality.

Federal level legislative changes are recommended to reduce the welfare disparities across the species. A single, uniform, federal law which regulates the rearing and slaughter of both bovines across all states has the potential to significantly improve the welfare of the dairy cows and buffaloes. Stricter regulations for owning either dairy bovine and the integration of smaller dairy farms into government-assisted co-operatives can lead to the provision of scientifically grounded, state-of-the-art practices for all bovines. The provision of dedicated bovine veterinary services to dairy farms, breeding centers, animal shelters, and slaughterhouses can notably alter the current welfare status and disparity. Finally, the promotion of buffalo farming as a religious endeavor based on the pre-Vedic Hindu beliefs, coupled with strict slaughter-permitting regulations can prove to be a means of welfare enhancement for the buffaloes.

### Limitations of the Study

Although this study pioneers in exploring the roles of socio-religious, economic, and legislative influences on the welfare of the Indian dairy bovines, it presents certain limitations. Its findings are limited to those obtained by utilizing the Five Freedoms framework. However, the conscious behavioral measures (such as the positive interactions that the animal has with humans) which form a measure of the Five Domains [20], are scarcely researched in multiple life phases of the bovines, and this lack of primary research should be taken into account before analyzing the welfare of the bovines through the more recent and comprehensive welfare measuring frameworks such as the Five Domains model. Although the majority of the Indian scientific literature are published in English [32], the review does not include Hindi literature surrounding dairy bovine welfare, limiting its analysis to literature published in English only.

## 5. Conclusions and Future Directions

India, the largest producer and consumer of milk [14,234], houses nearly one-third of the world’s cattle [1], with cows and buffaloes producing 97% of the total milk produced in the nation [2]. Evaluating the welfare of these bovines is essential especially in the light of the religiously influenced and widely prevalent cow slaughter ban. The inability to slaughter the bovine exerts a significant effect on the economics of the ubiquitous small-scale dairy farms which rear the majority of the nation’s dairy bovines [16,40,106], and thereby negatively impact the bovines’ welfare (The Cow Paradox). The literature reviewed demonstrates the loss of one or more of the Five Freedoms of animal welfare in multiple phases of the bovines’ productive lives, and a parallel loss of two or more of the freedoms in the post-productive lives of these animals, pointing to the role of religious values, legislation as well as the animal’s economic utility in the welfare of the bovines.

This study brings to light the largely ignored albeit crucial role of the culturally shaped human influences on the husbandry of the Indian dairy bovines and creates grounds for studying the human–bovine relationships through a holistic, interdisciplinary approach which takes into consideration both, the animal as well as the human element of the dynamic. Future research should examine the financial and the psychological effects of dairy farming on the preponderant small-scale dairy farmers. The motivations around cow protection should be re-evaluated by studying the cow protection sentiments across states that permit cow slaughter and those that do not. The effects of the bovine-related state legislation on the welfare of the bovines reared in that state should be examined. Indian Hindu dairy consumers’ awareness regarding the compromised welfare status of the dairy bovines and their opinions on the bovine-related legislation should be assessed. Exploring such meagerly researched areas can create a base for the generation of culturally viable solutions to improve bovine welfare and build a sustainable inter-species coexistence in the nation.

## Figures and Tables

**Figure 1 animals-15-00454-f001:**
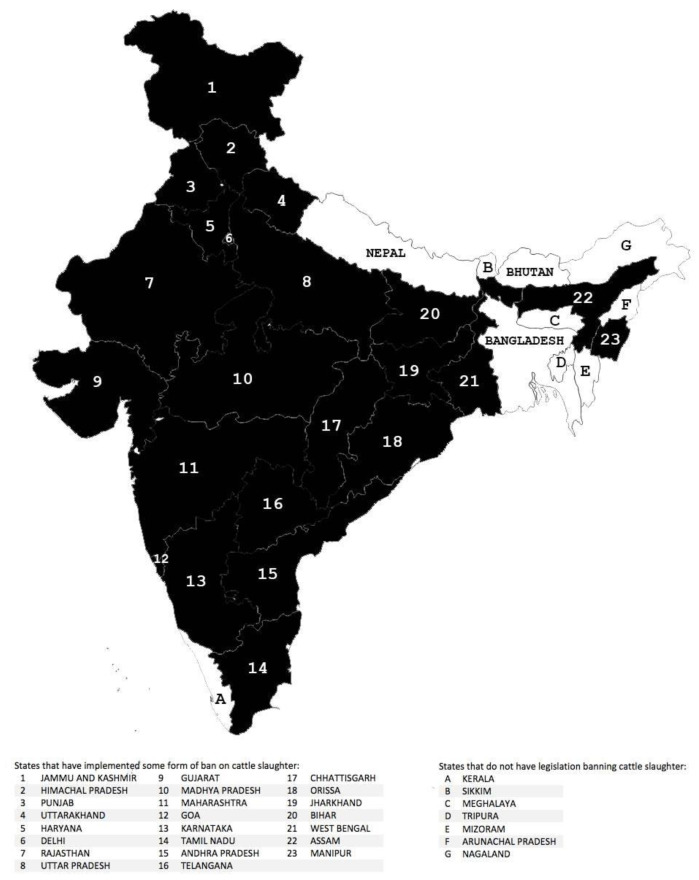
Political map of India with shaded areas indicating states that have implemented some form of ban on cattle slaughter [3]. Reproduced from Parikh & Miller, 2019. ACME: An International Journal for Critical Geographies, 18(4), 835–874, available under a Creative Commons Attribution 4.0 License.

**Figure 2 animals-15-00454-f002:**
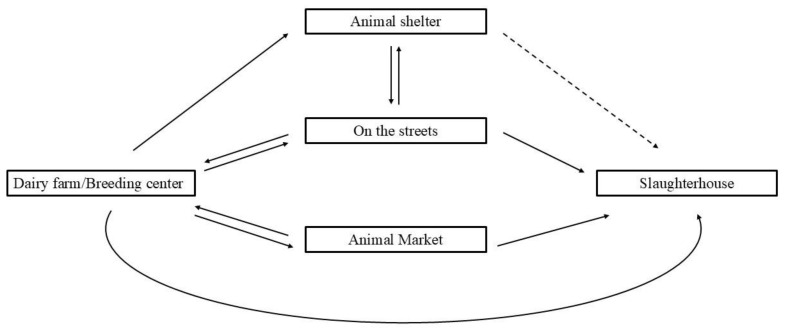
Phases in the lives of the Indian dairy bovines. Bovines on streets can be abandoned strays or farmer-owned/shelter-owned animals which are intermittently let out on the streets to fend food. Bovines may visit animal markets several times in their lives, from where they can be transported to either another dairy farm or sent for slaughter.

**Figure 3 animals-15-00454-f003:**
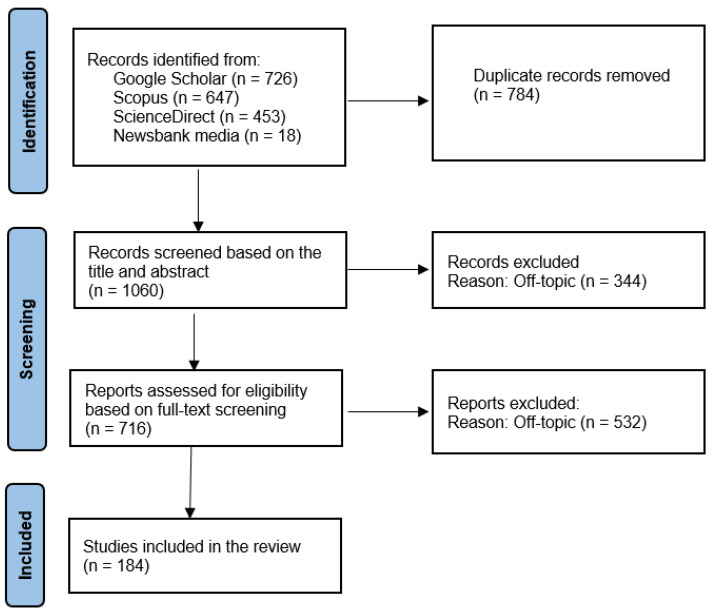
PRISMA (Preferred Reporting Items for Systematic reviews and Meta-Analyses) flowchart of the procedure for the selection of the studies analyzed in the review. The studies on Indian dairy bovine welfare were identified as observational, experimental, or secondary research on factors impacting welfare, governmental reports on welfare, or newspaper articles addressing welfare.

**Figure 4 animals-15-00454-f004:**
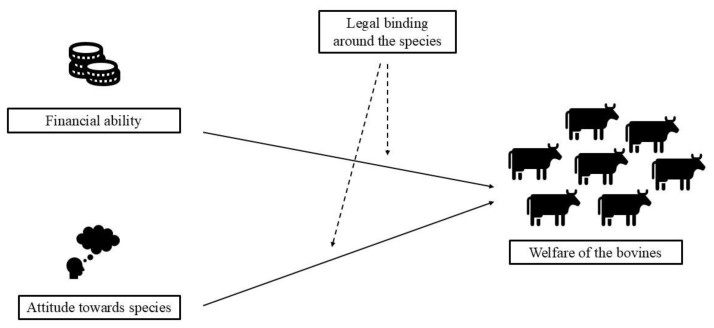
Determinants of bovine welfare in India. The animal rearer’s financial ability and attitude are known to impact the welfare of the dairy bovines. Given the widespread ban on cow slaughter in the nation, the financial ability and attitudes appear regulated by the extent of legalities surrounding the bovine species.

**Table 1 animals-15-00454-t001:** The Five Freedoms of animal welfare framework, entailing the provisions under each freedom. Adapted from the Farm Animal Welfare Committee’s 2009 report on farm animal welfare [19].

Freedom of Welfare	Provisions
Freedom from hunger and thirst	By ready access to water and a diet to maintain health and vigor
Freedom from discomfort	By providing an appropriate environment
Freedom from pain, injury, and disease	By prevention or rapid diagnosis and treatment
Freedom to express normal behavior	By providing sufficient space, proper facilities, and an appropriate company of the animal’s own kind
Freedom from fear and distress	By ensuring conditions and treatment which avoid mental suffering

**Table 2 animals-15-00454-t002:** Freedom-wise distribution of the peer-reviewed primary, peer-reviewed secondary and non peer-reviewed bibliographic references cited in the review, along with the time range of publication.

References Cited per Freedom	Peer-Reviewed Primary References	Peer-Reviewed Secondary References	Non Peer-Reviewed References	Publication Time Range
Freedom from hunger and thirst (n = 53)	26	17	10	1959–2023
Freedom from discomfort (n = 30)	21	5	4	1965–2023
Freedom from pain, injury, and disease (n = 58)	38	20	0	1981–2024
Freedom to express normal behavior (n = 17)	13	4	0	1978–2022
Freedom from fear and distress (n = 26)	22	2	2	1983–2023
Total references (n = 184)	120	48	16	

## Data Availability

The secondary data created in this study are available on request from the corresponding author.

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
