# Peer review of "The Cow Paradox—A Scoping Review of Dairy Bovine Welfare in India Using the Five Freedoms"

_animals, 2025, doi:10.3390/ani15030454_

Round 1
Reviewer 1 Report
Comments and Suggestions for Authors This study provides an interesting insight into the welfare of cows in India in the context of the country’s ban on cow slaughter. It is an extremely interesting approach that highlights the paradoxical, negative consequences of the widespread belief in the sanctity of these animals in Indian society. More specifically, the main objective of the article is to assess the welfare of Indian dairy cows at different stages of their lives through the prism of the five freedoms (freedom from hunger and thirst; freedom from discomfort; freedom from pain, injury and disease; freedom to express normal behavior; freedom from fear and distress). The undertaken research intention was fully implemented, and the greatest advantages of the work include: 1) innovative and interdisciplinary nature of the research: the author looks at the analyzed issues from different perspectives, combining knowledge and methods from various scientific disciplines (including law, economics, veterinary medicine, sociology, axiology); the procedure of presenting the cultural and socio-economic background of the analyzed issues should be highly appreciated, especially since its description contains numerous factual references; 2) methodological and structural correctness: the work contains a detailed description of the research methods used and the criteria for searching and selecting source material; the work is structurally correct, it is characterized by simplicity and clarity, the layout of the work is factual and logical (this part of the work can be treated as exemplary); 3) rich source documentation: for the purposes of the conducted research, a thorough review of available literature data was carried out, and the collected source material was used appropriately; the attached list of sources includes as many as 224 items of literature; 4) defining the research perspective: an extremely valuable part of the article is the point "5. Conclusions and Future directions" where, in addition to the substantive final conclusions, the research perspective is outlined by indicating specific problems that should be the subject of future scientific research. In conclusion, it should be stated that the reviewed study meets all the requirements for scientific works and can be published in the presented form. This will certainly contribute to the development of science and raising the level of social awareness in the subject of animal protection, which is one of the conditions for further civilizational progress.
Author Response
Comment 1: This study provides an interesting insight into the welfare of cows in India in the context of the country’s ban on cow slaughter. It is an extremely interesting approach that highlights the paradoxical, negative consequences of the widespread belief in the sanctity of these animals in Indian society. More specifically, the main objective of the article is to assess the welfare of Indian dairy cows at different stages of their lives through the prism of the five freedoms (freedom from hunger and thirst; freedom from discomfort; freedom from pain, injury and disease; freedom to express normal behavior; freedom from fear and distress). The undertaken research intention was fully implemented, and the greatest advantages of the work include: 1) innovative and interdisciplinary nature of the research: the author looks at the analyzed issues from different perspectives, combining knowledge and methods from various scientific disciplines (including law, economics, veterinary medicine, sociology, axiology); the procedure of presenting the cultural and socio-economic background of the analyzed issues should be highly appreciated, especially since its description contains numerous factual references; 2) methodological and structural correctness: the work contains a detailed description of the research methods used and the criteria for searching and selecting source material; the work is structurally correct, it is characterized by simplicity and clarity, the layout of the work is factual and logical (this part of the work can be treated as exemplary); 3) rich source documentation: for the purposes of the conducted research, a thorough review of available literature data was carried out, and the collected source material was used appropriately; the attached list of sources includes as many as 224 items of literature; 4) defining the research perspective: an extremely valuable part of the article is the point "5. Conclusions and Future directions" where, in addition to the substantive final conclusions, the research perspective is outlined by indicating specific problems that should be the subject of future scientific research. In conclusion, it should be stated that the reviewed study meets all the requirements for scientific works and can be published in the presented form. This will certainly contribute to the development of science and raising the level of social awareness in the subject of animal protection, which is one of the conditions for further civilizational progress.
Response 1: Thank you for your in-depth feedback. I appreciate your efforts to provide it. I too, am hopeful that this work will contribute to improving the lives of animals as well as humans.
Reviewer 2 Report
Comments and Suggestions for Authors
Overall Impression
I read your article with great interest. The manuscript is well-structured, providing a thorough narrative review of the welfare challenges faced by dairy bovines in India. Improving the welfare of dairy cattle in India represents an immense challenge and merits more research and debate, especially in the context of socio-religious, economic, and legislative frameworks. The study introduces the concept of "Cow Paradox," highlighting the cultural dichotomy between the treatment of cows and buffaloes, which adds originality to the discourse. However, the manuscript could benefit from greater conciseness in some sections and deeper critical analysis in others.
Level of English - The manuscript uses formal academic English suitable for a peer-reviewed journal. However, there are instances of repetitive discourse. Minor revisions to streamline language and improve clarity are recommended.
Title - I think the title is somewhat inaccurate. As it stands, it gives the idea that the slaughter ban is the main (or only) reason why cattle welfare in India is poor. On the other hand, the title, or the keywords, do not mention the Five Freedoms or what should be the main thesis of the manuscript: the cow paradox. Therefore, I suggest the following alternative: “The Cow Paradox – a narrative review of dairy bovine welfare in India using the Five Freedoms” or “The Cow Paradox – a narrative review of the Five Freedoms applied to dairy bovine welfare in India”
Keywords – Add Five Freedoms, buffalo, India
Introduction
The introduction effectively sets the stage by contextualizing the topic within India’s socio-economic and legislative landscape. In the objectives, you should clarify that the scope of your review of the welfare of “Dairy bovines” includes cows and buffaloes and add the species names (Linnaeus).
Methodology
I have a few comments regarding the methods that were used: a narrative review of the English literature under the five freedoms. Although I recognise that the narrative review approach lends itself well to explore the topic, especially given the diverse sources and limited primary research available, it also limits its impact and generalisation. It could be argued that a scoping review approach would be more appropriate, replicable and help decrease the risk of anecdotal selection of the literature.
You also restricted the search to the English language. India has two official languages; Hindi and English. I reckon that a considerable amount of research is published in Hindi. This absence must be mentioned.
It could be argued that the Five Freedoms is an outdated concept, mostly focused on negative welfare, and that other approaches that have gained traction, such as the Five Domains or the Welfare Quality principles and criteria, encompass a wider range of welfare dimensions. You do not provide a reason why you chose the Five Freedoms over more recent AW approaches.
Finally, more explicit detail on how sources were selected and analysed (e.g., specific inclusion/exclusion criteria) would strengthen the methodology. The information currently provided is insufficient to replicate the search. I recommend that you provide the full search instructions, including the number of hits, as supplementary material.
You also need to explain that the phases of the cows’ lives (dairy farms, breeding centers, animal shelters, streets, transportation, animal markets, slaughterhouse) will be used to structure the findings.
Results
The results are comprehensive and align well with the Five Freedoms framework. However, this chapter is extremely long and there are some overlaps between the sections. I also miss a birds-eye view of the retrieved references. Maybe it could benefit from a summary at the end to reinforce key findings. This could be achieved via a table with the total number of references per Freedom, differentiating between primary and secondary literature, peer-reviewed and non-peer reviewed, and the dates of references.
Discussion
I think the discussion needs some improvement. The discussion integrates findings effectively but could explore in more detail the implications of the results for policy and practice. The Cow Paradox is an interesting concept that should be expanded to include specific recommendations for addressing these welfare disparities. You also need to add a subsection with the limitations of the study, namely, the narrative approach, the lack of Hindi literature, and the welfare dimensions that could be undetected by using the 5 Freedoms.
Data Availability Statement - I do not agree with the Data Availability statement. In fact, new data was created and analysed in this study in the form of an Excel file. Data sharing is therefore applicable to this file which must be made available to other researchers, at least on reasonable request.
References
Given the large number of cited references, I was unable to double-check most of them. Please carefully review all the references before submission.
Author Response
Overall Impression
Comment 1: I read your article with great interest. The manuscript is well-structured, providing a thorough narrative review of the welfare challenges faced by dairy bovines in India. Improving the welfare of dairy cattle in India represents an immense challenge and merits more research and debate, especially in the context of socio-religious, economic, and legislative frameworks. The study introduces the concept of "Cow Paradox," highlighting the cultural dichotomy between the treatment of cows and buffaloes, which adds originality to the discourse. However, the manuscript could benefit from greater conciseness in some sections and deeper critical analysis in others.
Response 1: Thank you for your high-quality feedback. I have addressed each of your comments and have highlighted the changes made in the article. I believe your recommendations have significantly enhanced the quality of this article, and I appreciate your efforts and involvement in this.
Comment 2: Level of English - The manuscript uses formal academic English suitable for a peer-reviewed journal. However, there are instances of repetitive discourse. Minor revisions to streamline language and improve clarity are recommended.
Response 2: I believe most of the repetitive discourse occurs in the longest section of the article—the results section-- which analyzes the impacts of the common managemental practices across multiple freedoms. For example, the practice of calf weaning at day 0 with only brief suckling instances over the next days not only impacts the calf’s freedom from hunger and thirst, but also that from fear and distress and from disease. Sub-par housing infrastructure influences not only the bovines’ freedom from discomfort, but also impacts that from pain, injury and disease. Therefore, to improve clarity in the section, I have provided a summary of the findings at the end of the section per your suggestion to provide the reader with a bird’s eye-view while not depriving them of the details (lines 830-860).
Additionally, I have re-viewed the entire paper to check for grammatical errors, and rectified them to streamline the language and improve clarity.
Comment 3: Title - I think the title is somewhat inaccurate. As it stands, it gives the idea that the slaughter ban is the main (or only) reason why cattle welfare in India is poor. On the other hand, the title, or the keywords, do not mention the Five Freedoms or what should be the main thesis of the manuscript: the cow paradox. Therefore, I suggest the following alternative: “The Cow Paradox – a narrative review of dairy bovine welfare in India using the Five Freedoms” or “The Cow Paradox – a narrative review of the Five Freedoms applied to dairy bovine welfare in India”
Response 3: The title has been edited per the first suggestion. The word ‘narrative’ was replaced by the word ‘scoping’ per your suggestion to change the review approach in the Methodology section.
Comment 4: Keywords – Add Five Freedoms, buffalo, India
Response 4: The keywords recommended have been added.
Introduction
Comment 5: The introduction effectively sets the stage by contextualizing the topic within India’s socio-economic and legislative landscape. In the objectives, you should clarify that the scope of your review of the welfare of “Dairy bovines” includes cows and buffaloes and add the species names (Linnaeus).
Response 5: The scope of the review has been clarified by clearing stating both, the common names and the scientific names of the dairy bovines studied. (lines 131-133)
Methodology
Comment 6: I have a few comments regarding the methods that were used: a narrative review of the English literature under the five freedoms. Although I recognise that the narrative review approach lends itself well to explore the topic, especially given the diverse sources and limited primary research available, it also limits its impact and generalisation. It could be argued that a scoping review approach would be more appropriate, replicable and help decrease the risk of anecdotal selection of the literature.
Response 6: Modifications have been made to the methods to utilize the scoping review approach by following the PRISMA guidelines. I have added details on the search instructions, such that the study avoids the risk of anecdotal selection of literature and becomes replicable (lines 182-199). A rationale for using this method has been provided at the beginning of the Materials and Methods section (lines 171-176).
Comment 7: You also restricted the search to the English language. India has two official languages; Hindi and English. I reckon that a considerable amount of research is published in Hindi. This absence must be mentioned.
Response 7: India does have two official languages, English and Hindi. However, given a lack of terminology to describe scientific phenomena and principles, majority of the scientific education and research in the nation takes place in English. I have mentioned this absence of referring to Hindi articles in the article selection criteria (lines 192-195).
Comment 8: It could be argued that the Five Freedoms is an outdated concept, mostly focused on negative welfare, and that other approaches that have gained traction, such as the Five Domains or the Welfare Quality principles and criteria, encompass a wider range of welfare dimensions. You do not provide a reason why you chose the Five Freedoms over more recent AW approaches.
Response 8: The rationale for choosing the Five Freedoms framework over other more recent ones has now been provided in lines 143-150.
Comment 9: Finally, more explicit detail on how sources were selected and analysed (e.g., specific inclusion/exclusion criteria) would strengthen the methodology. The information currently provided is insufficient to replicate the search. I recommend that you provide the full search instructions, including the number of hits, as supplementary material.
Response 9: Specific details on how sources were selected and analyzed are now written in the methodology section (lines 182-199, 227-233). I have also provided a flowchart to illustrate the selection process of the studies analyzed (Figure 3) and described it in lines 214-219.
Comment 10: You also need to explain that the phases of the cows’ lives (dairy farms, breeding centers, animal shelters, streets, transportation, animal markets, slaughterhouse) will be used to structure the findings.
Response 10: The explanation of the structuring of the results has now been provided at the end of the methods section (lines 231-233).
Results
Comment 11: The results are comprehensive and align well with the Five Freedoms framework. However, this chapter is extremely long and there are some overlaps between the sections. I also miss a birds-eye view of the retrieved references. Maybe it could benefit from a summary at the end to reinforce key findings. This could be achieved via a table with the total number of references per Freedom, differentiating between primary and secondary literature, peer-reviewed and non-peer reviewed, and the dates of references.
Response 11: A summary sub-section has been created to add a brief summary of the key findings (lines 830-860). I have also added a table entailing the freedom-wise distribution of the references, along with the specific peer-reviewed/non peer-reviewed, and primary/secondary details (Table 2.)
Discussion
Comment 12: I think the discussion needs some improvement. The discussion integrates findings effectively but could explore in more detail the implications of the results for policy and practice. The Cow Paradox is an interesting concept that should be expanded to include specific recommendations for addressing these welfare disparities. You also need to add a subsection with the limitations of the study, namely, the narrative approach, the lack of Hindi literature, and the welfare dimensions that could be undetected by using the 5 Freedoms.
Response 12: I have expanded on the implications of the results for policy and practice, with a focus on two major welfare influences—legislation and religious beliefs. I have added these implications immediately following the interpretation of the results (lines 894-913).
Based on the concept of the Cow Paradox, I have included specific recommendations that address the welfare disparities that exist between the two species. These recommendations have been added as a paragraph following those elaborating the Cow Paradox (lines 947-957).
A limitations sub-section has been added which details the study’s limitations: the lack of Hindi literature, and the welfare dimensions that could be undetected by using the 5 Freedoms. (lines 959-967) Based on your suggestions, I had edited the review approach to that of a scoping review, such that it follows the PRISMA guidelines. Therefore, the narrative review approach is no longer applicable as a limitation, and has not been listed in this sub-section.
Comment 13: Data Availability Statement - I do not agree with the Data Availability statement. In fact, new data was created and analysed in this study in the form of an Excel file. Data sharing is therefore applicable to this file which must be made available to other researchers, at least on reasonable request.
Response 13: The data availability statement has been edited per your suggestion (lines 998-999)
References
Comment 14: Given the large number of cited references, I was unable to double-check most of them. Please carefully review all the references before submission.
Response 14: Each reference was double-checked using Zotero, the same referencing software that was used to initially collect, organize and cite the work referred to in the article.

Round 2
Reviewer 2 Report
Comments and Suggestions for Authors
Thank you for accommodating my comments and suggestions.
I have only minor comments:
You haven’t included Bos indicus as a target species. I realise that most of the literature does not indicate whether by cow they mean Bos indicus or Bos taurus, but I am pretty sure that a significant portion of Indian livestock are Bos indicus breeds like Gir, Sahiwal, and Red Sindhi. Most of the cattle is probably crossbred, but a reference to Bos indicus needs to be included.
I think you need to improve the justification for using the Five Freedoms. You use a post-hoc fallacy to justify its use. How can you tell that “the literature available will not be able to provide adequate inputs required to satisfactorily measure certain parameters of these frameworks” without trying? You know that after having conducted the analysis, not before! You need to rephrase it as a hypothesis. You should probably say that the Five Freedoms provide an objective and measurable framework to explore a previously unexplored topic and evidence suggests that this framework has been successfully applied in similar contexts to assess animal welfare, making it a suitable choice for this study. Then the reflection that “conscious behavioral measures (such as the positive interactions that the animal has with humans) which form a measure of the Five Domains [20], are scarcely researched in multiple life phases of the bovines” (lines 147-149) should be moved to the discussion (thus soothing the limitations of the study).
Line 144 – Include a reference for the Five Domains framework and the Welfare Quality protocol, respectively.
I suggest moving the Brief summary of the findings to the beginning of the results section. It makes more sense for the reader to have a bird-eye view of the results before reading them in detail.
Delete comma (,) on line 849.
Reference 55 – Change from uppercase to sentence case.
Well done!
Author Response
Thank you for accommodating my comments and suggestions.
I have only minor comments:
Comment 1: You haven’t included Bos indicus as a target species. I realise that most of the literature does not indicate whether by cow they mean Bos indicus or Bos taurus, but I am pretty sure that a significant portion of Indian livestock are Bos indicus breeds like Gir, Sahiwal, and Red Sindhi. Most of the cattle is probably crossbred, but a reference to Bos indicus needs to be included.
Response 1: I agree with your statement, and have now made a reference to Bos indicus on line 132.
Comment 2: I think you need to improve the justification for using the Five Freedoms. You use a post-hoc fallacy to justify its use. How can you tell that “the literature available will not be able to provide adequate inputs required to satisfactorily measure certain parameters of these frameworks” without trying? You know that after having conducted the analysis, not before! You need to rephrase it as a hypothesis. You should probably say that the Five Freedoms provide an objective and measurable framework to explore a previously unexplored topic and evidence suggests that this framework has been successfully applied in similar contexts to assess animal welfare, making it a suitable choice for this study. Then the reflection that “conscious behavioral measures (such as the positive interactions that the animal has with humans) which form a measure of the Five Domains [20], are scarcely researched in multiple life phases of the bovines” (lines 147-149) should be moved to the discussion (thus soothing the limitations of the study).
Response 2: I agree with your comments. Edits have now been made in lines 143-149 and lines 955-960 per your suggestions.
Comment 3: Line 144 – Include a reference for the Five Domains framework and the Welfare Quality protocol, respectively.
Response 3: References for the Five Domains framework and the Welfare Quality protocol have now been included.
Comment 4: I suggest moving the Brief summary of the findings to the beginning of the results section. It makes more sense for the reader to have a bird-eye view of the results before reading them in detail.
Response 4: Thank you for suggesting this. The Brief summary of the findings has now been moved to the beginning of the results section (lines 235-259).
Comment 5: Delete comma (,) on line 849.
Response 5: The comma has now been deleted (now line 248).
Comment 6: Reference 55 – Change from uppercase to sentence case.
Response 6: The reference (now reference # 61) has been changed from upper to sentence case.
Comment 7: Well done!
Response 7: Thank you! And thank you for reviewing my work.
